# Remote Sensing Imagery Super Resolution Based on Adaptive Multi-Scale Feature Fusion Network

**DOI:** 10.3390/s20041142

**Published:** 2020-02-19

**Authors:** Xinying Wang, Yingdan Wu, Yang Ming, Hui Lv

**Affiliations:** 1School of Science, Hubei University of Technology, No. 28 Nanli Road, Wuhan 430068, China; xiny_wang@163.com (X.W.); lvhui@hbut.edu.cn (H.L.); 2Hubei Collaborative Innovation Centre for High-Efficient Utilization of Solar Energy, Hubei University of Technology, No. 28 Nanli Road, Wuhan 430068, China; 3Center for Spatial Information Science and Systems, George Mason University, Fairfax, VA 22030, USA; 4Hubei Engineering Technology Research Center of Energy Photoelectric Deviceand System, Hubei University of Technology, No. 28 Nanli Road, Wuhan 430068, China; 5Institute of Surveying and Mapping, CCCC Second Highway Consultants Co., Ltd, No. 18 Chuangye Road, Wuhan 430056, China

**Keywords:** super-resolution, remote sensing imagery, adaptive multi-scale feature fusion

## Abstract

Due to increasingly complex factors of image degradation, inferring high-frequency details of remote sensing imagery is more difficult compared to ordinary digital photos. This paper proposes an adaptive multi-scale feature fusion network (AMFFN) for remote sensing image super-resolution. Firstly, the features are extracted from the original low-resolution image. Then several adaptive multi-scale feature extraction (AMFE) modules, the squeeze-and-excited and adaptive gating mechanisms are adopted for feature extraction and fusion. Finally, the sub-pixel convolution method is used to reconstruct the high-resolution image. Experiments are performed on three datasets, the key characteristics, such as the number of AMFEs and the gating connection way are studied, and super-resolution of remote sensing imagery of different scale factors are qualitatively and quantitatively analyzed. The results show that our method outperforms the classic methods, such as Super-Resolution Convolutional Neural Network(SRCNN), Efficient Sub-Pixel Convolutional Network (ESPCN), and multi-scale residual CNN(MSRN).

## 1. Introduction

Image super-resolution (SR), is a classical yet challenging problem in the field of computer vision. The goal of image super-resolution is to reconstruct a visually pleasing high-resolution (HR) image from one or more low-resolution (LR) images [1]. Remote sensing imagery, captured from the satellite optical imaging sensors, provides abundant information to monitor the Earth’s surface, having broad application in the fields of object matching and detection, land cover classification, assessment of urban economic levels, resource exploration, etc. It has proved that high-resolution remote sensing images play an important role. However, due to factors such as long-distance imaging, atmospheric turbulence, transmission noise, and motion blurring, the quality and the spatial resolution of remote sensing imagery are relatively poorer and lower as compared with natural images. Moreover, the ground objects of remote sensing imagery usually have different scales, causing the objects and surrounding environment to mutually couple in the joint distribution of their image patterns [2]. Therefore, super-resolution for remote sensing imagery has attracted huge interest and become a hot research topic.

The methods of image super-resolution can be divided into two groups: Multiple Image Super-Resolution (MISR) [3] and Single Image Super-Resolution (SISR) [4,5,6]. The former requires a set of LR images to reconstruct an HR image. SISR is more popular in practice, and the method proposed in this article belongs to the SISR. 

To solve the problem of SISR, various algorithms, such as interpolation-based, reconstruction-based, and learning-based methods have been developed over the past few decades. In recent years, due to the rapid development of deep learning theory, the deep learning-based super-resolution methods have gradually become mainstream [7]. Super-Resolution Convolutional Neural Network (SRCNN) is the first deep learning algorithm for image super-resolution [8]. It treats each step of sparse coding as a convolutional neural network process, and directly establishes an end-to-end mapping between a pair of LR image patch and a HR one. However, it consumes much time for the input of the network a bicubic interpolation version of a LR image. Fast Super-Resolution Convolutional Neural Network (FSRCNN) [9] is the upgraded version with higher calculation efficiency, as it can input the original low-resolution image to the network directly by using deconvolution layer, and this makes the dimension of the input to the network relatively small, beneficial to reduce the computation and accelerate the reconstruction speed. In a similar way, Shi et al. [10] proposed Efficient Sub-Pixel Convolutional Network (ESPCN). Generally, the effective size of image context for reconstruction is correlated with the receptive field size of CNN, which depends on the convolutional kernel size in each layer and the depth of CNN. However, the above methods are essentially shallow networks, and they may have problems, such as small receptive field and insufficient image information for image reconstruction.

To solve this problem, Lin et al. proposed a dilated CNN to enlarge receptive field size for the image super-resolution [11]. Kim et al. [12] proposed Very Deep Super-resolution Convolutional Networks (VDSR) with 20 layers to obtain large receptive field, and learned only the residuals between the LR image and the HR image to accelerate the convergence speed. Tai et al. [13] proposed Deep Recursive Residual Network(DRRN), which arranged the basic residual units and blocks in a recursive topology to increase the depth of network and reduce parameter number. An increasing number of methods used the intermediate feature information of the network to improve the performance, such as DRCN [14], SRResNet [15], SRDenseNet [16], and MemNet [17]. Huang et al. [18] established a dense convolutional network (DenseNet), instead of passing the features of the previous layer to the next layer sequentially, and features extracted from all previous layers were passed to the current layer. Li et al. [19] proposed a multi-scale residual CNN (MSRN), which introduced multi-scale convolutional filters to enhance the feature inference capability. Xu et al. [20] proposed a global dense feature fusion convolutional network (DFFNet) for single image super-resolution of different scale factors, in which cascaded feature fusion blocks were used to learn global features in both spatial and channel direction. Deeper and more complicated networks are developed for image super-resolution.

Nowadays, many scholars apply CNN for the reconstruction of remote sensing imagery [21]. Lei et al. [22] proposed a multi-fork structure to extract local and global features of remote sensing images and obtain good reconstruction results. Xu et al. [23] used deep memory connection to combine image details of remote sensing images with environmental information. Jiang et al. [24] designed an ultra-dense residual network that uses the rich long- and short-line connections in the network to enhance the network’s ability to extract remote sensing image features. Gu et al. [25] added a squeeze and excitation (SE) module to the network to improve the network representation. Dong et al. [26] used enhanced residual block and residual channel attention group to obtain multi-level remote sensing feature information. Lu et al. [2] reconstructed high-resolution remote sensing images by extracting patches of different sizes as multi-scale information input networks and fusing high-frequency information of different scales. However, the problem of redundant feature information is often ignored. In addition, the network structure for feature extraction and fusion is often fixed in above networks. Adaptive feature information extraction and fusion would be better for the remote sensing imagery super-resolution, due to complex factors of image degradation and diversity of image content.

For remote sensing image super-resolution, this paper proposes an Adaptive Multi-scale Feature Fusion Network (AMFFN). AMFFN can extract dense features directly from the original low-resolution image without any image interpolation preprocessing. Several adaptive multi-scale feature filtering blocks are cascaded to adaptively extract high-frequency detailed feature information of remote sensing imagery.

In summary, this paper contributes the following:(1)An adaptive multi-scale feature fusion network for the remote sensing image super-resolution, which can adaptively extract multi-scale feature information;(2)The mechanisms of squeeze-and-excited and adaptive gating are integrated for feature extraction and fusion, which can learn the channel correlation of feature maps, adaptively decide how much of the previous feature information should be reserved, reduce the redundant feature information among the intermediate multi-scale feature and enhance the use of useful feature information.

The remainder of this article is organized as follows. In Section 2, the network structure and the implementation details are discussed in detail. Section 3 demonstrates the experimental results on remote sensing image super-resolution, and the comparisons with other classical methods are discussed. The conclusions are given in Section 4.

## 2. Adaptive Multi-Scale Feature Fusion Network

### 2.1. Network Architecture

The network structure of AMFFN consists of four parts: Original feature extraction, adaptive multi-scale feature extraction, feature fusion and image reconstruction, as shown in Figure 1 and the part of adaptive multi-scale feature extraction is the core of our algorithm.

The input of our network is the original low-resolution image for sup-resolution, denoted as $I_{LR}$. A convolutional layer conv with $n_{0}$ filters are firstly applied to the input image to produce a set of feature maps,
(1)$$A_{0}=w_{0}*I_{LR}+b_{0}$$
where $A_{0}$ is the original feature maps extracted from the low-resolution remote sensing imagery, $w_{0}$ corresponds filters in the convolutional layer, which is 128 filters with the spatial size of 3 × 3 in this paper, $b_{0}$ denotes the biases of the convolutional layer, and ‘∗’ represents the convolution operation. 

In the part of adaptive multi-scale feature extraction, supposing there are n adaptive multi-scale feature extraction (AMFE), and the output of i-th AMFE $A_{i}$ can be represented as,
(2)$$A_{i}=f_{MFE}(A_{i-1})+g\left(A_{i-1}\right)\;(1\leq i\leq n)$$
where $f_{MFE}(\cdot )$ denotes the operation of multi-scale feature extraction, and g(⋅) represents adaptive feature gating operation, the details will be elaborated in the following sub-section. AMFE is the basic module for the adaptive feature extraction, which consists of a unit of multi-scale feature extraction (MFE) and a feature gating for adaptively retaining the feature information from the output of previous AMFE. 

Through feature extraction, a series of feature maps, such as $A_{0},\cdots ,A_{n}$, can be obtained. These feature maps contain a large amount of redundant information, which increase the computational burden significantly if they are directly used for image reconstruction. Therefore, before delivering these feature for super-resolution, a feature fusion layer is stacked after AMFE for feature fusion and reduction. The output of feature fusion layer $A_{fusion}$ is formulated as,
(3)$$A_{fusion}=w_{f}*[A_{0},A_{1},\cdots ,A_{n}]+b_{f}$$
where $w_{f}$ corresponds to the weights of the feature fusion layer, which is 64 filters of a size of 1 × 1, $b_{f}$ is the corresponding biases, and $\left[A_{0},A_{1},\cdots ,A_{n}\right]$ denotes the concatenation of all feature maps extracted by the first feature extraction layer conv and AMFE. 

As many CNN-based SISR methods, the sub-pixel convolution method is adopted to reconstruct the high-resolution image. The reconstruction function can be defined as follows,
(4)$$I_{SR}=w_{s2}*shuffle(w_{s1}*A_{fusion})$$
where $w_{s1}$ denotes the weights of a 3 × 3 convolution layer. If the scale factor is r(e.g., ×2), the number of filters in the convolution layer would be $C\cdot r^{2}$, and C refers to the channel number of the input feature maps.shuffle(⋅) represents the shuffling operation that rearranges the elements of a $H_{LR}\times W_{LR}\times C\cdot r^{2}$ tensor acquired in the top layer into a $rH_{LR}\times rW_{LR}\times C$ tensor, more details can consult to [10]. A 3 × 3 convolution layer $w_{s2}$ with $C_{1}$ filters used to reconstruct the remote sensing images, and $C_{1}$ represents the number of channels of the original input image (e.g., if it is an RGB image, $C_{1}=3$). And the tensor of $rH_{LR}\times rW_{LR}\times C_{1}$ is our desired reconstructed high-resolution image $I_{SR}$. In our paper, L1 function is chosen to avoid introducing unnecessary training tricks and reduce computations.

### 2.2. Adaptive Multi-Scale Feature Extraction

As previously mentioned, the module of adaptive multi-scale feature extraction (AMFE) is the core module in our method. The structure of AMFE is illustrated as Figure 2, and it mainly consists of two units: Multi-scale feature extraction and filtering, and feature gating. For multi-scale unit feature extraction and filtering, it contains two parts: Multi-scale feature extraction and feature filtering.

#### 2.2.1. Multi-Scale Feature Extraction Unit

Firstly, the feature maps outputted from the previous AMFE are processed with a convolutional layer. For the i-th AMFE, this can be defined as,
(5)$$M_{i}^{0}=\phi (w_{i}^{0}*A_{i-1}+b_{i}^{0})$$
where $A_{i-1}$ is the feature map from the previous AMFE, the number of feature maps outputted from each AMFE is 128 in the paper, $w_{i}^{0}$ corresponds to 128 filters of a size of 128×3×3, $b_{i}^{0}$ is the corresponding biases, and ϕ(⋅) represents activation function Relu. 

Then, three types of filters $f_{i_{1}}=1\times 1$, $f_{i_{2}}=3\times 3$ and $f_{i_{3}}=5\times 5$ is used to extract multi-scale features, the numbers of these filters are all 64, which can be expressed by Equation (6),
(6)$$M_{i}^{1j}=\phi (w_{i}^{1j}*M_{i}^{0}+b_{i}^{1j})\;(j=1,2,3)$$
where j denotes the type index of the filters. Suppose that each filter bank contains $n_{i_{1}}=n_{i_{2}}=n_{i_{3}}=64$, and the convolutional output is concatenated and divided into 3 groups, that is $\left[M_{i}^{0},M_{i}^{11},M_{i}^{12}\right]$, $\left[M_{i}^{0},M_{i}^{11},M_{i}^{13}\right]$, $\left[M_{i}^{0},M_{i}^{12},M_{i}^{13}\right]$, as shown in Figure 2. Then, three different 1×1 convolution layers $\left\{w_{i}^{21},w_{i}^{22},w_{i}^{23}\right\}$ with 64 filters each are utilized to learn the channel correlation between the extracted multi-scale features of each group. The output feature maps $\left\{M_{i}^{21},M_{i}^{22},M_{i}^{23}\right\}$ are then concatenated to $\left[M_{i}^{21},M_{i}^{22},M_{i}^{23}\right]$, and a 1×1 convolution layer $w_{i}^{3}$ with 256 filters is used again to further extract the feature information of all. These process can be expressed by the following,
(7)$$\{\begin{array}{c} M_{i}^{21}=w_{i}^{21}*\left[M_{i}^{0},M_{i}^{11},M_{i}^{12}\right]+b_{i}^{21}\\ M_{i}^{22}=w_{i}^{22}*\left[M_{i}^{0},M_{i}^{11},M_{i}^{13}\right]+b_{i}^{22}\\ M_{i}^{23}=w_{i}^{23}*\left[M_{i}^{0},M_{i}^{12},M_{i}^{13}\right]+b_{i}^{23} \end{array}$$
(8)$$M_{i}^{3}=w_{i}^{3}*\left[M_{i}^{21},M_{i}^{22},M_{i}^{23}\right]+b_{i}^{3}$$

With filters of different spatial size and the cascaded structure of AMFE, we can build a hierarchical system that can extract multi-scale image feature information. The filter of spatial size 1 × 1 is mainly used to perform dimension reduction, but it also can learn the channel correlation between the feature maps, that is “extract” feature information along the channel direction.

#### 2.2.2. Feature Filtering Unit

To enhance the sensitivity of informative features, feature filtering follows the multi-scale feature extraction. We borrowed the idea of squeeze-and-excitation (SE), proposed by the Hu et al. [27], to promote useful features and suppress less useful ones. The SE method firstly used global average pooling to generate channel-wise statistics, which was used as a channel descriptor. Then, two fully-connected (FC) layers around the non-linearity are used to form a bottleneck to derive the scalar corresponding each feature map. For high computation efficiency, the FC layers are replaced by the 1 × 1 convolution layer, and the diagram of feature filtering is illustrated in Figure 3. The operation of feature filtering unit can be defined as follows,
(9)$$M_{i}^{4}=w_{i}^{4}*\left(A_{impor}(M_{i}^{3})\times M_{i}^{3}\right)+b_{i}^{4}$$
where $A_{impor}(\cdot )$ represents the operation of determining the importance score of each feature map, $w_{i}^{4}$ corresponds to 128 filters with spatial size of 1 × 1.

#### 2.2.3. Feature Gating Unit

When the structure of the network is fixed, it would be non-adaptation and not flexible enough to copy with the complex situation, especially for the remote sensing imagery. Therefore, in the module of adaptive multi-scale feature extraction, a simple feature gating mechanism is adopted in this paper, as illustrated by Figure 1. A shortcut connection enables the features outputted from the previous AMFE module to feed to current AMFE module directly. This is beneficial for reducing the loss of feature information during the transmission. In this paper, a feature gating mechanism is used to adaptively decide how much of the previous feature information should be reserved, and the implementation details are shown in the Figure 4. 

The key for feature gating is how to adaptively obtain the value of gating score $score(A_{i-1})$ for the input feature $A_{i-1}$. When the value of gating score is determined, which is a scalar, then the reserved feature information ${A^{\prime }}_{i-1}$ is just as follows,
(10)$${A^{\prime }}_{i-1}=g(A_{i-1})=score(A_{i-1})\times A_{i-1}$$
where g(⋅) represents the gating operation. To calculate the gating score and alleviate the calculation burden, the average pooling is used to reduce the dimension of the feature map, and use the global information to learn the gating score. Then, to capture the dependencies between channels, we add a simple non-linear function of two fully-connected layers connected with BatchNorm [28] and a ReLU activation function, and the output is a vector V of two elements. After softmax operation, vector V would be a normalized vector with V[0]+V[1]=1. We define the second element V[1] is our desired value of gating score, which represents the how much proportion of feature information need to be reserved. 

To enhance the robustness, the noise with Gumbel distribution is added when deriving the vector V, that is the Gumbel-Softmax strategy [29] is used to replace the softmax. Then, the new vector $V^{\prime }$ is calculated as follows,
(11)$$V^{\prime }=softmax((V+G)/\tau )$$
where G is the Gumbel noise vector, each element $G_{i}$ follows Gumbel(0,1) distribution, which can be sampled using inverse transform sampling by drawing $u_{i}∼$ Uniform(0,1) and computing $G_{i}=-\log(-\log(u_{i}))$, τ is the softmax temperature, which is set to 1 in our paper.

## 3. Experimental Results and Analysis

### 3.1. Datasets and Performance Metrics

To verify the effectiveness of the method in this paper, three datasets of remote sensing imagery are used. The first is the UC Merced land-use dataset (referred to as the UC later) [30]. The dataset includes 21 types of scenes, and each scene has 100 images with size of 256 × 256 pixels and spatial resolution of 0.3 m. For each type of scene, 80 images are randomly selected into the training image set, and the remaining 20 images are selected into the test image set. The second is NWPU-RESISC45 (referred to as NW later) [28]. The dataset contains 45 types of scenes, each of which has 700 images with the same size of 256 × 256 pixels and the spatial resolution varying from 30 meters to 0.2 meters. For each type of scene, 100 images are randomly selected into the training image set, and 10 images are randomly selected from the remaining into the test image set. The third is the images captured by the satellite TianGong-2 (referred to as the TG later) [31]. The dataset consists of 6 types of scenes, and the total number of images is 2000, which are all selected into the test image set. Through these, we build our training image set and testing image set. The overall information and the example images of the experimental datasets are illustrated in Table 1 and Figure 5 respectively.

The algorithm is based on the PyTorch framework, which enables NVIDIA TitanXp Graphics Processing Unit (GPU)and Intel (R) Xeon (R) Silver 4116 Central Processing Unit (CPU) to train the model. The original high-resolution images are downsized by bicubic interpolation to generate corresponding low-resolution images for training, and the training images are augmented by horizontal or vertical flipping and 90° rotating transformation. For all training images, low-resolution patches with a size of 64 × 64 are extracted, and the total number of LR image patches is 11,124. In each training batch, we randomly extract 16 LR patches with the size of 64 × 64 and an epoch having 696 iteration of back-propagation. The maximum epoch number is 100, the learning rate is 0.0001, and the Adam optimizer is used. The peak signal-to-noise ratio (PSNR) and structural similarity (SSIM) are selected as the metrics for the evaluation of each experiment.

### 3.2. Network Analysis

#### 3.2.1. Number of AMFE Modules

AMFE module is the core part of our network, and its number dramatically affects the depth of our network. To find suitable value for it, the number of AMFE is set to 2, 4, 8, 16, 24, and 32, and the values of Loss and PSNR are depicted in Figure 6 and Table 2. 

From Table 2, it can be found that the PSNR of reconstructed images increases with the number of AMFE modules. This can be explained by that with the increase of the number of AMFEs, more feature information could be extracted, which is beneficial for the super-resolution of remote sensing imagery. The increase rate is gradually slow down with the number of AMFEs. And when the number of AMFEs reaches to 16, our network reaches the best performance, as shown in Figure 6, the loss value decreases faster and is more stable than others, and the PSNR is the highest. When the number of AMFEs is 24 or 32, the network costs more time and has a relatively slow convergence, but a worse result. The overfitting and the insufficient training data may be the reason for this. For our experiment, we set the number of AMFEs to be 16 considering the tradeoff between the performance and computing efficiency.

#### 3.2.2. Adaptive Feature Gating

To verify the effectiveness of our feature gating mechanism, three connection methods with the MFE are discussed, which are: 1)The output of MFE is directly used as the input of the next MFE;2)Add a shortcut to connect the output of previous MFE with the input of the next MFE;3)Replace the shortcut of the way 2) with our gating mechanism.

The comparison results are given in Figure 7 and Table 3. We can see that by adding the skip connection, it enables to directly learn the difference between the features and reaches a faster convergence speed, and with our gating mechanism, it shows better performance on both the convergence rate and the PSNR value. The convergence speed is faster than the short cut connection way after the 40 epochs and the PSNR is higher than other two methods by about 0.3 dB. The skip connection provides a shortcut to connect the output of previous MFE directly with the input of the next MFE, which is beneficial to the propagation of feature information, but it may result in information redundancy. In addition, excessive parameters might lead to overfitting. This maybe the reason that the skip connection method achieves worse result. Our feature gating strategy can learn from the practical images and adaptively determine the gating score, which decides how much proportion of feature information from previous MFE will be reserved and integrated. From the experimental results, we can find that the feature gating unit can reduce redundant information effectively and improve the performance of image super-resolution.

### 3.3. Comparision Results with Other Classical Methods

Our proposed method AMFFN has been compared with classical methods, such as Bicubic interpolation, SRCNN [8], ESPCN [10], and MSRN [19]. The quantitative results of these methods for scale factor ×2, ×3, and ×4 are in Table 4. To ensure fairness, SRCNN, ESPCN, MSRN and our network AMFFN are trained and tested by the same remote sensing image set.

Compared with SRCNN and ESPCN, the PSNR obtained by our method is higher by 3 dB to 5 dB, a significant improvement has been achieved. The reason for this is that our method can extract multi-scale feature and realize adaptive feature fusion, which contributes to the enhancement of results of image super-resolution. However, SRCNN and ESPCN are essentially shallow networks, with limiting ability of feature extraction and fusion. When contrasting to the MSRN method, which is also a deep network and achieves better results than SRCNN and ESPCN, our method outperforms it in terms of PSNR and SSIM. 

Visual comparisons on scale factor ×2 are shown in Figure 8, Figure 9, Figure 10, Figure 11 and Figure 12. From the results, it is found that AMFFN can clearly reconstruct the green plants and the straight strips in the farm field in UC dataset. However, the methods of Bicubic interpolation, SRCNN and ESPCN cannot accurately reconstruct the green plants. MSRN can reconstruct some green plants, but the ringing effects arises when reconstructing the straight strips in the farmland. For the images of urban scene in the NW dataset and mountain and farmland scenes in the TG datasets, the linear features and spatial structure of reconstructed high-resolution images are clearer using our method.

## 4. Conclusions

This paper proposes an adaptive multi-scale feature fusion network for remote sensing imagery. Several adaptive multi-scale feature extraction (AMFFN) are used to extract multi-scale feature information, and the squeeze-and-excited and feature gating unit mechanism are adopted to enhance the adaptation of feature information, to adaptively select and make full use of intermediate feature information. Quantitative and visual benchmarking results on different test data sets show that our AMFFN outperform the classical image super-resolution methods. 

## Figures and Tables

**Figure 1 sensors-20-01142-f001:**
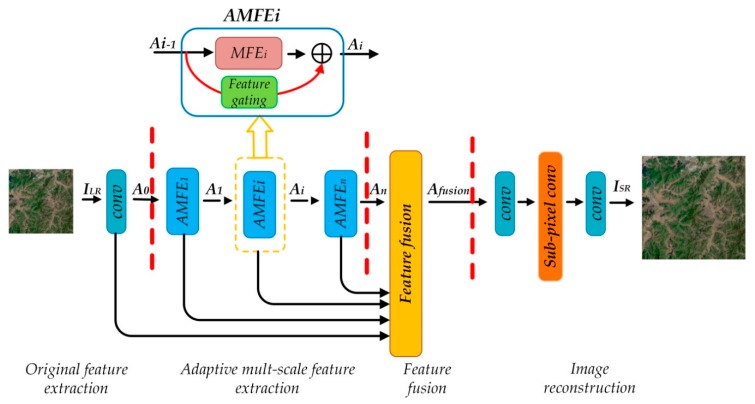
Network architecture of the proposed method.

**Figure 2 sensors-20-01142-f002:**
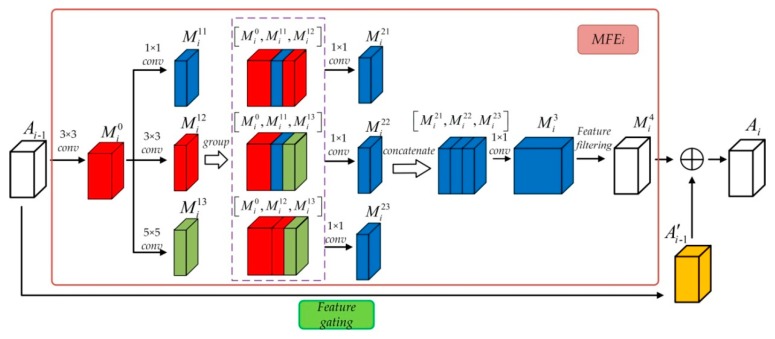
The structure of AMFE module.

**Figure 3 sensors-20-01142-f003:**
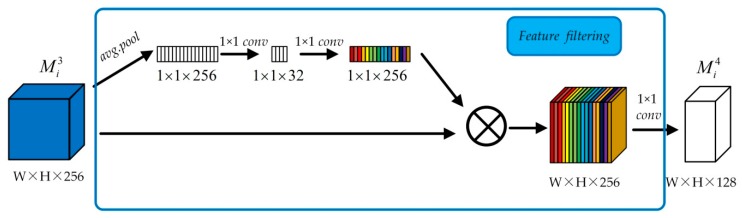
Diagram of feature filtering.

**Figure 4 sensors-20-01142-f004:**
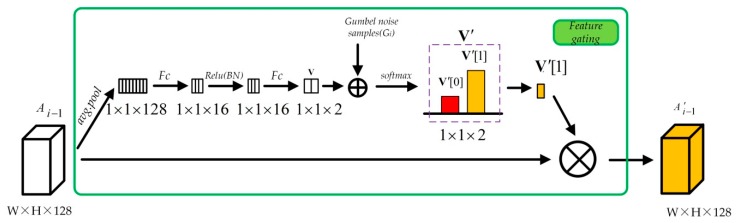
Schematic diagram of feature gating unit.

**Figure 5 sensors-20-01142-f005:**
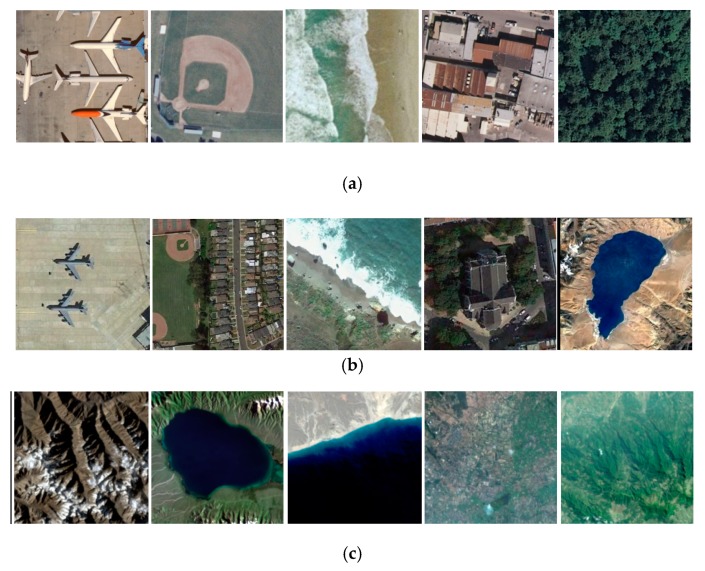
Example images of three datasets: (**a**) UC dataset, (**b**) NW dataset, (**c**) TG dataset.

**Figure 6 sensors-20-01142-f006:**
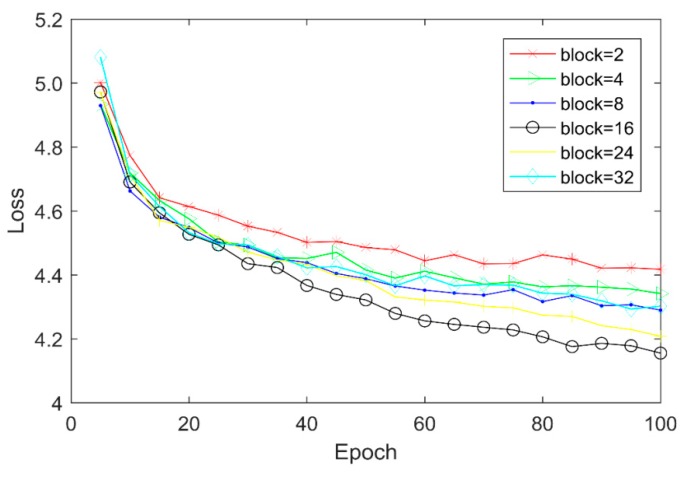
Loss values of different numbers of AMFE with scale factor ×2.

**Figure 7 sensors-20-01142-f007:**
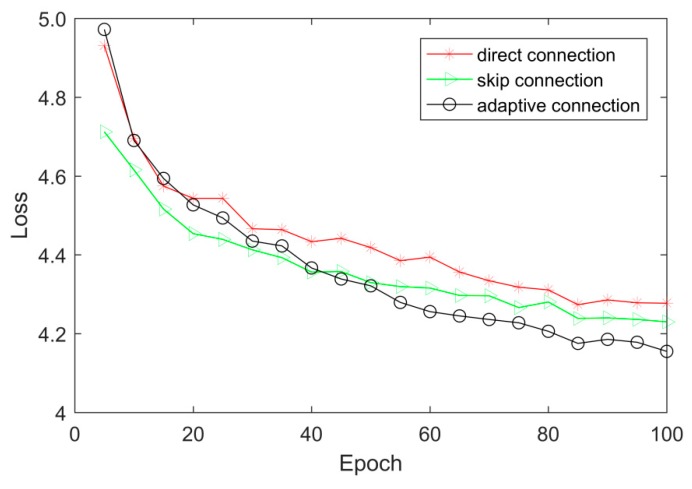
Loss values with different connection methods with scale factor ×2.

**Figure 8 sensors-20-01142-f008:**
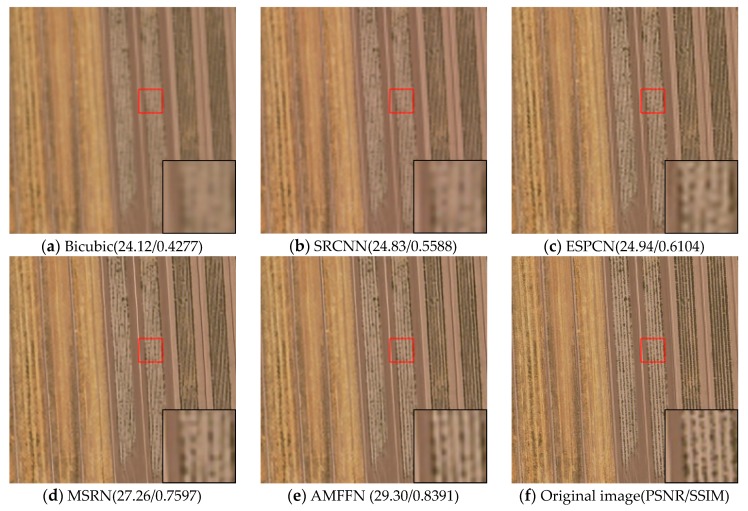
Results of farmland scenes in UC dataset of different methods with scale factor ×2.

**Figure 9 sensors-20-01142-f009:**
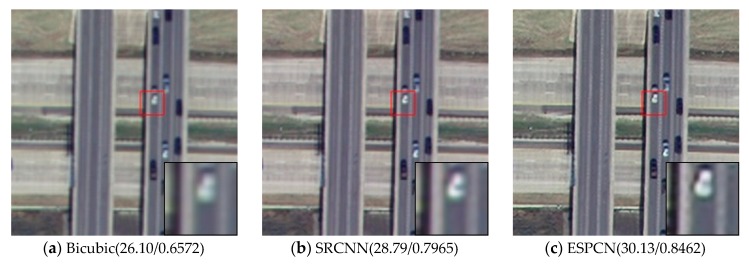
Results of road scenes in UC dataset of different methods with scale factor ×2.

**Figure 10 sensors-20-01142-f010:**
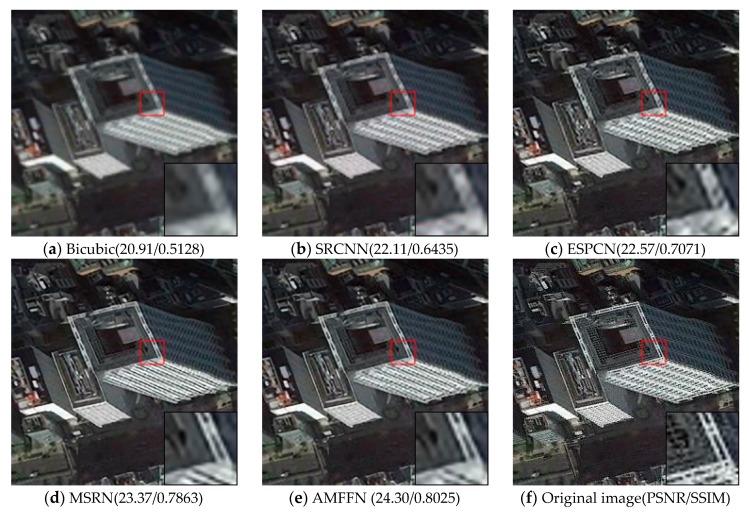
Results of building scenes in NW dataset of different methods with scale factor ×2.

**Figure 11 sensors-20-01142-f011:**
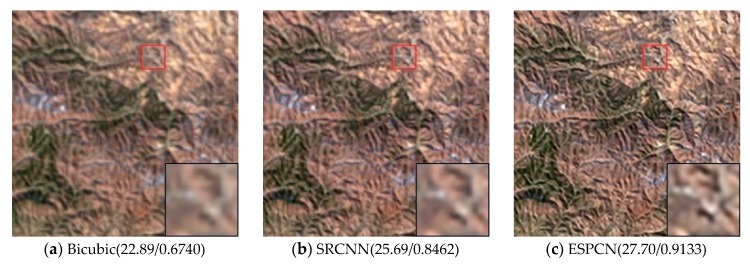
Results of mountain scenes in TG dataset of different methods with scale factor ×2.

**Figure 12 sensors-20-01142-f012:**
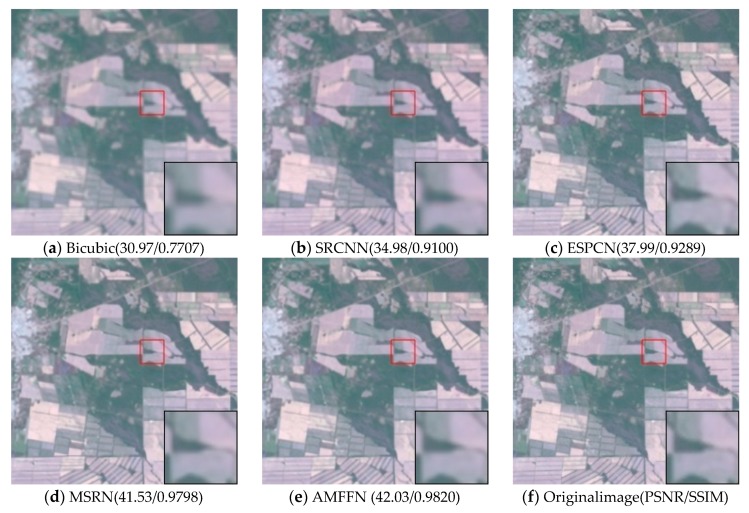
Results of farmland in TG dataset of different methods with scale factor ×2.

**Table 1 sensors-20-01142-t001:** Information of experimental data.

Dataset	Scene Number	Image Size (Pixels)	Image Number	Training Image Number	Testing Image Number	Spatial Resolution (m)
UC	21	256 × 256	2100	1680	420	0.3
NW	45	256 × 256	31,500	4500	450	0.2~30
TG	6	256 × 256	2000	0	2000	100

**Table 2 sensors-20-01142-t002:** PSNR and cost time of different numbers of AMFE with scale factor ×2.

Number of AMFEs	2	4	8	16	24	32
PSNR(dB)	38.92	39.01	39.04	39.76	39.43	39.24
time(s)	0.12	0.14	0.17	0.24	0.31	0.37

**Table 3 sensors-20-01142-t003:** PSNR and cost time consumption with different connection methods with scale factor ×2.

Connection Methods	1	2	3
PSNR(dB)	39.49	39.39	39.76
time(s)	0.20	0.20	0.24

**Table 4 sensors-20-01142-t004:** Comparison results with other classical methods.

Datasets	Scale Factor	Bicubic	SRCNN	ESPCN	MSRN	AMFFN
		PSNR/SSIM	PSNR/SSIM	PSNR/SSIM	PSNR/SSIM	PSNR/SSIM
**UC**	×2	26.75/0.6402	29.67/0.7842	31.51/0.8184	34.83/0.9342	35.00/0.9360
	×3	24.32/0.4734	26.40/0.5842	27.70/0.6625	30.80/0.8539	30.94/0.8581
	×4	22.72/0.3402	24.72/0.4408	25.60/0.5221	28.61/0.7718	28.70/0.7772
NW	×2	27.94/0.6687	30.83/0.8156	32.46/0.8455	34.93/0.9296	35.30/0.9348
	×3	25.53/0.4977	27.71/0.6264	28.87/0.6967	31.32/0.8465	31.37/0.8477
	×4	23.95/0.3599	26.08/0.4804	26.85/0.5580	29.42/0.7746	29.47/0.7763
TG	×2	32.13/0.7219	35.48/0.8629	37.22/0.8477	40.40/0.9678	40.55/0.9682
	×3	29.32/0.5617	31.49/0.6974	33.10/0.7357	35.79/0.9062	35.84/0.9067
	×4	27.50/0.4185	29.52/0.5492	30.70/0.6115	33.34/0.8413	33.36/0.8420

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
