# Peer review of "Remote Sensing Imagery Super Resolution Based on Adaptive Multi-Scale Feature Fusion Network"

_sensors, 2020, doi:10.3390/s20041142_

Round 1

Reviewer 1 Report

The proposal of an adaptive network of fusion of characteristics at multiple scales (AMFFN) is very appropriate, as well as the idea of balancing the depth of the learning network and the efficiency of the computer process, and especially suggestive, is to pass the characteristics extracted from all layers before the current layer.

The workflow is very well defined and the proposed and validated solution allows you to extract features of multiple scales and perform adaptive fusion functions, both very convenient aspects to reconstruct high resolution images.

Author Response

Thank you to the reviewers for your approval of our manuscript.

We checked the manuscript again and fixed some typographical and grammatical errors.

We hope that the corrections can be received smoothly.

Reviewer 2 Report

This manuscript uses an adaptive multi-scale feature fusion network to get a super-resolution image from a low-resolution remote sensing image. The proposed method seems to perform well, but the explanations are not clear enough.

1.Are the number n in line 120 and the number n in line 127 the same?
2.How to determine the parameters w^0 and b^0 in Eq. (1)? Or, what are their values?
3.How to determine the parameters w^f and b^f in Eq. (3)? Or, what are their values?
4.How to determine the parameter w^s in Eq. (4)? Or, what are the values?
5.How to do the operation "shuffle" in Eq. (4)?
6.How to determine the parameters w's and b's in Eq. (5)? Or, what are their values?
7.How to determine the parameters w's and b's in Eq. (6)? Or, what are their values?
8.In line 209, what is the equation or curve for the ReLU function used by this manuscript?
9.In line 214, the vector V is used to replace the softmax. But, the softmax is still used in line 217.
10.In line 219, what are u^i?
11.In line 220, what is the value of "tau" used in this manuscript?
12.In line 241, SSIM is the abbreviation for "structural similarity", not for "structural similarity index".
13.What is the parameter "epoch" in Figs. 6 and 7?
14.The method MSRN in line 292 is not introduced in Sec. 1.
15.For Ref. [37], the No. 1709.0150 is wrong.
16.The formats for the author names, pages, paper names, and journal or conference names of the references are in chaos a little.
17.There are many grammar errors, e.g. the errors in lines 36, 65, 101, 103-107, 184, 198, 255-257, 264, 272-273, 275, 288, 295, 298,...

In conclusion, extra detailed explanations and corrections for grammar errors are needed. Only one revision is allowed.

Reviewer 3 Report

"Remote Sensing Imagery Super Resolution Based on Adaptive Multi-scale Feature Fusion Network"

An adaptive multi-scale feature fusion network for remote sensing image super-resolution has been proposed in this paper. Features are extracted from the original low-resolution image, then several adaptive multi-scale feature extraction (AMFE) blocks are applied, the extracted features are fused and finally the sub-pixel convolution method is used to reconstruct the high-resolution image. The adaptive multi-scale feature extraction modules incorporate squeeze-and-excited and adaptive gating mechanisms.

Several adaptive multi-scale feature filtering blocks are cascaded to adaptively extract high-frequency detailed feature information.
Features extracted in previous AMFE blocks are conserved after applying an adaptive feature gating operation to select only non-redundant features.

It is shown that the proposed method outperforms state of the art methods super-resolving remote sensing images.

Some errata:

- Paper's reference 24 has been published in IEEE Transactions on Multimedia.
- Lines 262-263: "... as show in the Fig.7 ...". It should refer to Fig.6.
- A minor English text revision has to be performed.

Author Response

Point 1:Paper's reference 24 has been published in IEEE Transactions on Multimedia.

Response 1: Suggestions are accepted. Thank you for your correction. Reference 24 has been modified and you can view it in P13L384 of the manuscript.

Point 2:Lines 262-263: "... as show in the Fig. 7 ...". It should refer to Fig. 6.

Response 2: Suggestions are accepted. This error was corrected in P8L175.

Point 3:A minor English text revision has to be performed.

Response 3: Suggestions are accepted, and we have checked and corrected grammatical errors in the manuscript.

Reviewer 4 Report

Since the complexity of remote sensing image super-resolution, the authors proposed an adaptive multi-scale feature fusion network (AMFFN) which is a redesigned version of MSRN. According to the reported results, the proposed AMFFN outperforms several methods, however, the research and the manuscript are not well organized. The reviewer think this work is not suitable for publication. Some issues are as follows:

Point 1: References [2-8] are not related to remote sensing image super-resolution. References [13-23] are less related to the work reported in this paper. Discussions about the outstanding SR methods in remote sensing community and the most recent state-of-the-art SR methods are missing completely.

Point 2: Some descriptions are inconsistent. P1L31: The goal of image super-resolution is to ...... from ‘a’ low-resolution image input. P1L42: The methods can be classified into: ‘multiple’ image super-resolution and single image super-resolution.

Point 3: As described in the abstract, the proposed AMFFN directly extracts features from low-reoslution image and uses sub-pixel convolution to construct high-resolution image. These are widely employed by the CNN-based SR methods since 2017.

Point 4: P3L101: An adaptive multi-scale feature fusion network which can adaptively fuse the feature information. If the feature fusion corresponds to the third part of AMFFN, it belongs to the contributions of MSRN, I think.

Point 5: This paper is really difficult to read. The settings of the convolutional layers need to be clarified. What are the channel numbers of the layers? The mechanism of the feature filtering needs to be better described.

Point 6: P5L171: One concatenation group which contains M0, M11, M12, and M13 is more efficient than three, I think. P5L175: A convolutional layer of kernel size 1x1 can be used to extract information? Are you sure?

Point 7: Why remote sensing image super-resolution is more complex? Why SISR is more efficient than MISR? (P1L44) Why shallow networks have small receptive field and can not extract features sufficiently? (P2L72) How did the filters implement extracting multi-scale features? (P5L165) How did the channel number reduce from 256 to 128? (P5Figure3) Which loss function is employed? How about the performance of AMFFN with 24 AMFEs? What are the corresponding super-resolving factors of Figue 6, Figure 7, Table 2, and Table 3? What are the evaluation scores of the results shown in Figure 9-12? These are not well presented.

Point 8: P9L282: Adding skip connections enables the network to learn residuals directly and reach a faster convergence. Why adding skip connections led to a lower PSNR value (as reported in Table 3)? Some naive discussions, such as the effectiveness of the feature gating unit mentioned at P9L286, are not acceptable to readers.

Point 9: The proposed AMFFN was trained and evaluated with images from same scene types. This is unreasonable. If MSRN was not trained with remote sensing images, the comparisons in this paper are not objective.

Point 10: Spell check is required. P3L101: can adaptively fusing. P3L119: sup-resolution.

Round 2

Reviewer 2 Report

The revised manuscript is ready to be published.

Author Response

Thank you to the reviewers for your approval of our manuscript.

We checked the manuscript again and fixed some formatting and syntax errors. We hope that the manuscript will be accepted smoothly.

Reviewer 4 Report

Please see my comments in the attached file.
